# The Effect of Using Participatory Working Time Scheduling Software on Employee Well-Being and Workability: A Cohort Study Analysed as a Pseudo-Experiment

**DOI:** 10.3390/healthcare9101385

**Published:** 2021-10-16

**Authors:** Rahman Shiri, Kati Karhula, Jarno Turunen, Aki Koskinen, Annina Ropponen, Jenni Ervasti, Mika Kivimäki, Mikko Härmä

**Affiliations:** 1Finnish Institute of Occupational Health, 00250 Helsinki, Finland; kati.karhula@ttl.fi (K.K.); Jarno.Turunen@ttl.fi (J.T.); Aki.Koskinen@ttl.fi (A.K.); annina.ropponen@ttl.fi (A.R.); jenni.ervasti@ttl.fi (J.E.); mika.kivimaki@helsinki.fi (M.K.); mikko.harma@ttl.fi (M.H.); 2Division of Insurance Medicine, Department of Clinical Neuroscience, Karolinska Institute, 171 77 Stockholm, Sweden; 3Department of Public Health, University of Helsinki, 00014 Helsinki, Finland

**Keywords:** propensity score, psychological distress, self-rated health, self-rostering, work-life conflict, worktime control

## Abstract

Shift workers are at increased risk of health problems. Effective preventive measures are needed to reduce the unfavourable effects of shift work. In this study we explored whether use of digital participatory working time scheduling software improves employee well-being and perceived workability by analysing an observational cohort study as a pseudo-experiment. Participants of the Finnish Public Sector cohort study with payroll records available between 2015 and 2019 were included (N = 2427). After estimating the propensity score of using the participatory working time scheduling software on the baseline characteristics using multilevel mixed-effects logistic regression and assigning inverse probability of treatment weights for each participant, we used generalised linear model to estimate the effect of using the participatory working time scheduling software on employees’ control over scheduling of shifts, perceived workability, self-rated health, work-life conflict, psychological distress and short sleep (≤ 6 h). During a 2-year follow-up, using the participatory working time scheduling software reduced the risk of employees’ low control over scheduling of shifts (risk ratio [RR] 0.34; 95% CI 0.25–0.46), short sleep (RR 0.70; 95% CI 0.52–0.95) and poor workability (RR 0.74; 95% CI 0.55–0.99). The use of the software was not associated with changes in psychological distress, self-rated health and work-life conflict. In this observational study, we analysed as a pseudo-experiment, the use of participatory working time scheduling software was associated with increased employees’ perceived control over scheduling of shifts and improved sleep and self-rated workability.

## 1. Introduction

Work at social and healthcare organisations is organised in several shifts to provide 24-h service [1]. Studies suggest that shift workers are at increased risk of sleep disturbance [1,2], poor mental health [1,3,4] and work-life conflict [5]. Night shift workers have been found to sleep less and report more often fatigue, insomnia and mental health problems than day workers [6], with more marked differences between shift and day workers observed in younger age groups [7,8]. There is a need to identify practical ways to reduce the unfavourable effects of shift work.

Worktime control has beneficial effect on work-non-work balance [9], and is defined as control over starting and ending of workdays, length of shifts, taking breaks and holidays and timing of overtime [9]. Theoretical groundings for the assumed benefits of increased work-time control stem from several motivational and occupational health theories [10,11]. Worktime control and self-scheduling of shifts have been suggested to improve well-being [12] and have also other beneficial health effects [13]. In a recent cross-sectional study adjusted for the propensity score, self-scheduling of work shifts among nurses was associated with better organisational justice and work attitude [14]. In prospective observational cohort studies, poor worktime control has been associated with increased risk of psychological distress [15,16] and depressive symptoms [17,18]. Unpredictable and irregular work schedules have also been linked to poor sleep quality [16,19] and poor mental health [4]. In addition, work-life conflict has been more common in employees who work > 40 h per week [5] or have short intervals (<11 h) [5] between work shifts. Conversely, employees with a flexible schedule seem to sleep better [20] and report less often work-life conflict [21] or poor self-rated health [20] than workers without a flexible schedule. In our earlier observational study, employees using the participatory working time scheduling software reported increased control over scheduling of shifts [22], and reduced rates of excessive sleepiness [22] and sickness absence [23]. However, another study of self-scheduling software found no evidence on decreased stress among elderly care workers [24] and an intervention to increase control over working time did not confirm improvements in sleep quality [25].

Observational studies are prone to selection bias and confounding [26]. However, confounding in prospective cohort studies can be limited with rigour data analysis. The propensity score methods, for example, are commonly used to analyse prospective cohort studies as pseudo-trials [27], studies intended to assess causal relationship without random assignment. The propensity score estimates the probability of receiving a treatment, an intervention or an exposure conditional on measured baseline characteristics [27]. The estimated propensity score is then used to achieve balance in background characteristic across treatment groups through matching, weighting or stratification [27]. A well conducted propensity score weighting is more efficient to control confounding in observational studies than regression adjustment [27,28].

In the current study, we analyzed an observational cohort study as a pseudo-trial to explore whether using the participatory working time scheduling software improves shift workers’ well-being and perceived workability. In addition, we examined whether beneficial health effects of using the participatory working time scheduling software differ between younger and older employees.

## 2. Methods and Material

### 2.1. Population

This study is part of ongoing Working Hours in the Finnish Public Sector study [22] based on payroll data of working hours, and comprised of employees who worked in five hospitals districts and one division of municipal health services in Finland between 2015 and 2019. A total of 5207 hospital employees with payroll data on working hours were included in 2015 survey, 6080 employees included in 2017 survey and 4920 included in 2019 survey (Figure 1). We used two employee cohorts, one with baseline in 2015 and follow-up in 2017 and the other with baseline in 2017 and follow-up in 2019. From both cohorts, we excluded day workers, employees who worked <31 days in past 3 months, worked <150 days in past year, or physicians who worked on-call shifts > 90 days in past 91 days or >90 days in past year. There were 2224 employees in the 2015–2017 cohort and 1912 employees in the 2017–2019 cohort. In total these provided 4136 observations, of which 1709 had missing data on baseline characteristics and were not included in the propensity score. After pooling the two cohorts, there were 2427 employees at study baseline. Of these, 881 were users and 1546, non-users of the participatory working time scheduling software. The study outcomes were measured at follow-up, two years after the baseline. A total of 402 participants had missing data on one or more study outcomes at follow-up and were excluded from the main analysis. Thus, the final analytic sample consisted of 2025 employees, including 881 users and 1144 non-users of the participatory working time scheduling software with data on the baseline characteristics and the outcomes of interest. This pseudo-experiment was registered on ClinicalTrials.gov (NCT02775331) before initiation of the intervention.

### 2.2. Outcomes

The study outcomes were control over scheduling of shifts, perceived workability, self-rated health, psychological distress, short sleep and work-life conflict. *Control over scheduling of shifts* was assessed with a single Likert question: ‘How much control do you have over the scheduling of work shifts?’ The response alternatives included (1) very much, (2) quite a lot, (3) some, (4) quite a little and (5) very little. We defined intermediate control as having some control over scheduling of shifts (response 3) and defined low control as having quite a little or very little control over scheduling of shifts.

A Likert scale question from the Work Ability Index [29] was used to measure *workability*. The responses ranged from 0 (unable to work at all) to 10 (ability to work is at its best). The single question is strongly associated with the Work Ability Index and both showed similar patterns of associations with health outcomes [29]. We used a dichotomised outcome and defined poor workability as score ≤ 6. *Self-rated heath* was assessed with a single-item ‘How do you rate your health?’ and response alternatives included (1) good, (2) fairly good, (3) average, (4) fairly bad and (5) poor. The question is widely used and recommended as an indicator of health in surveys [30]. We dichotomised fairly bad or poor vs. others. The 12-item General Health Questionnaire (GHQ-12) was used to assess *psychological distress*. The responses to each item range from 0 to 3 (coded 0-1-2-3). We recoded all items as 0-0-1-1 [31], and defined psychological distress as score ≥ 3 [32]. Sleep duration was inquired with a question ‘How many hours do you usually sleep during a 24-h period?’. We defined *short sleep* as sleep duration ≤6 h. *Work-life conflict* was assessed using a single question: how often do you feel that your work takes too much time or energy from your family or life? The response alternatives were (1) never, (2) rarely, (3) sometimes, (4) often and (5) very often. We dichotomised often/very often vs. others.

### 2.3. Use of the Participatory Working Time Scheduling Software

The intervention group consisted of hospital employees who used the participatory working time scheduling software [23]. The participatory scheduling software allows employees interactively schedule the shifts. After negotiations and alterations, the head nurse accepts the roster for a three-week period. The participatory scheduling enables the employees to influence their working time and enter their desired shifts into their wards’ shift schedule following collectively agreed rules on, e.g., the number of night shifts or proportion of weekends off-work. The employees are also able to see their co-workers who will be working the same shift. The hospitals internally decided which wards and when to start using the participatory working time scheduling software. The hospitals also made all decisions about the length of introduction period and training whereas the rules for shift scheduling were made at ward level.

The control group consisted of hospital employees who used traditional scheduling (from here on non-users of the software for brevity). In the traditional working time scheduling, the head nurse schedules the shifts for a three-week period at least two weeks prior to the start of the period. The employees have limited influence on their working time. The head nurse makes the final decisions on the final schedules in all cases. We have applied intention-to-treat principle and included all participants who had been using the software for at least a month.

### 2.4. Baseline Characteristics

To facilitate a rigorous propensity score weighting, a wide range of baseline data on characteristics potentially affecting the study outcomes and selection into the user vs. non-user group of the software were collected. These included personal and work-related factors. Detailed description of some measures is reported in Appendix A Table A1.

*Personal factors:* Information on age, sex, marital status, consumption of beer, wine or other low-alcohol drinks per week, consumption of spirit per month, smoking status (never, past, current), self-reported height and weight, number of children and history of stressful life events was gathered in both 2015 and 2017 surveys. Information on education, vocational training and history of medical conditions was collected in 2015 survey. History of medical conditions included allergy, asthma, bronchitis, hypertension, heart disease, cerebrovascular disease, osteoarthritis, rheumatoid arthritis, low back pain, sciatica, peptic ulcer, migraine, depression, other mental disorders, diabetes, high cholesterol level and sleep apnoea. For 2017–2019 cohort, we utilised data on these background characteristics collected in year 2015 survey.

Leisure-time physical activity was assessed with four questions in 2015 and 2017 surveys. Information on the employees’ average weekly hours of leisure-time physical activity within the past 12 months was collected regarding four grades of intensity: (1) walking, (2) brisk walking, (3) jogging, and (4) running, or their equivalent activities [33]. The number of hours per week for each activity grade ranged between zero and four hours. A metabolic equivalent (MET) index was calculated for each participant by multiplying the MET-values of each activity intensity by the time spent on them, and summing [34,35]. We used tertile distribution and split the MET index into low, medium and strenuous activity.

*Work-related factors:* Information on the types of usual work shifts, type of work time (part-time vs. full-time), number of days of on-call in a month and working as a supervisor was gathered in both 2015 and 2017 surveys. A question on types of usual shift had four alternative responses: (1) shift work without night shifts (two-shift work), (2) shift work with night work (three-shift work), (3) regular night work and (4) other irregular work. The Job Content Questionnaire was used to assess job demands (3 items) and job control (9 items). In all the 12 items, response alternatives ranged from 1 (strongly disagree) to 5 (strongly agree) and total scores were computed for job demands and job control after reversing 2 items for job control. To create job strain for each participant, job demands and control were dichotomised using the median distribution and job strain defined as experiencing high demands combined with low control [36]. Procedural justice was assessed by seven-item [37]. The items assess perceived fairness of managerial procedures, and responses ranged from 1 (strongly agree) to 5 (strongly disagree). We used tertile distribution and split the sum score into high, intermediate and low justice levels.

The magnitude and significance of the change at work was measured with a single item and responses ranged from 1 (changes have been small and insignificant) to 7 (changes have been large and significant). We used tertile distribution and split the changes into small, medium and large. Employee’s involvement in planning changes at work was also assessed with a single item and responses included (1) I can influence change very much, (2) I can influence some and (3) change usually comes unexpectedly without my ability to influence it.

Uncertainty at work was assessed with five Likert scale questions about the threat of termination of some jobs, transfer to other tasks, forced redundancies, dismissal and increase in workload beyond tolerance. Rewards for work in the forms of income, benefits, appreciation and satisfaction was assessed with four Likert scale questions. Responses to the questions on uncertainty and rewards at work ranged from 1 (very much) to 5 (very little). We summed the responses and included the sum scores as continuous variables in the propensity score models. The participants were also asked about fatigue during working hours or leisure time, their intention to retire early and discrimination at workplace on the grounds of age, sex, education, opinion, ethnicity, sexual orientation.

### 2.5. Statistical Analysis

We used the total sample of the 2015–2017 cohort (N = 5539 employees) and assessed intraclass correlation coefficient for hospital wards. Intraclass correlation coefficient measures similarities of responses within clusters and it was 0.206 for control over scheduling of shifts, 0.082 for self-rated health, 0.051 for work conflict, 0.038 for workability, 0.024 for psychological distress and 0.013 for short sleep at follow-up.

We estimated the propensity score of using the participatory working time scheduling software on the baseline characteristics (i.e., year 2015 survey for the 2015–2017 cohort and year 2017 survey for the 2017–2019 cohort, Figure 1) using multilevel mixed-effects logistic regression. We used three-level random-intercept model of using the participatory working time scheduling software on baseline characteristics with individuals nested within wards and wards nested within hospitals. We included baseline characteristics as well as outcome variables at baseline. Baseline characteristics in the propensity scores were: age (continuous variable), sex, education (three levels), vocational training (three levels), marital status, being a supervisor, type of work time, types of usual shifts, on-call work in a month (dichotomised), changes at work, employee’s involvement in changes at work, uncertainty at work, work rewards, intention to retire, discrimination, history of medical conditions (12 conditions), leisure time physical activity (three levels), body mass index (continuous variable), having a child, smoking status (never, past, current), alcohol consumption, history of stressful life events (illness or death), high job demands, job strain, procedural justice (three levels), fatigue during working hours or leisure time, psychological distress (GHQ-12), workability, control over scheduling of shifts, work-life conflict, duration of sleep and self-rated health.

We created an inverse probability of treatment weight for each participant using the propensity score and assigned weight to employees based on the inverse of their probability of using the participatory working time scheduling software. We stabilised the inverse probability of treatment weights to reduce the variability and bias [38]. We then used generalised linear model with a binomial distribution and a log link function. This method allows analytical weights (aweights in Stata) for inverse probability of treatment weight. Some of the participants in the control group were included in both cohorts and we controlled for repeated observations using *vce* (*cluster)* option. As a sensitivity analysis, we estimated subgroup balancing propensity scores for employees aged <50 years and those aged ≥50 years [39]. We assessed whether weighting balanced the baseline characteristics between users and non-users of the participatory working time scheduling software [27]. We estimated the unweighted and weighted standardised differences in baseline characteristics to compare prevalence and means between users and non-users. We used Stata, version 17 for the analyses.

## 3. Results

The baseline characteristics for the total study population and by age groups are reported in Table 1. Of the 2427 participants at baseline, 90% were women and 55.8% had completed at least upper secondary school education. The job titles with 50 or more participants included nurse (36.2%), practical nurse (9.5%), department secretary (6.8%), supply/instrument technician (4.3%), radiology nurse (4.1%), midwife (3.5%), specialist (2.5%) and mental health nurse (2.1%). Sixty-eight percent of employees aged <50 years and 50% of employees aged ≥50 years worked three shifts (including nights). Low control over scheduling of shifts was more common in employees aged ≥ 50 years (29.2%) than employees aged < 50 years (18.8%). This was also the case for short sleep (19.5% in employees aged ≥50 years vs. 11.4% in those aged <50 years). Thirty percent of employees aged <50 years and 28.2% of employees aged ≥50 years has psychological distress, and 7.1% of employees aged <50 years and 15.6% of employees aged ≥50 years reported poor workability at baseline.

Table 2 shows prevalence of the outcomes at follow-up. Intermediate or low control over scheduling of shifts, poor perceived workability, poor self-rated health and short sleep were more prevalent among employees aged ≥ 50 years, whereas work-life conflict and psychological distress were more prevalent among employees aged <50 years.

Comparison of baseline characteristics between users and non-users.

Balance in the baseline characteristics between users and non-users was achieved for most of the baseline characteristics (Table 3). Of the 43 baseline characteristics included in propensity score, standardised difference between users and non-users was 10% or higher for 15 variables in unweighted sample and for 5 variables in the weighted sample. In the weighted sample, standardised difference was 10% or higher for education, vocational training, part-time job, types of shifts and control over scheduling of shifts. Weighting reduced imbalances between users and non-users at baseline by 31% for education and by 43–52% for vocational training, part-time job, types of shifts and control over scheduling of shifts.

Associations of using participatory working time scheduling software with the outcomes.

Employees who used participatory working time scheduling software had more control over scheduling of their shifts than non-users at follow-up (Table 4). The risk of low control over scheduling of shifts was three-fold lower in users than non-users (risk ratio [RR] 0.34, 95% CI 0.25–0.46). Moreover, users were at lower risk of poor workability (RR 0.74, CI 0.55–0.99) and short sleep (RR 0.70, CI 0.52–0.95) than non-users. Using participatory working time scheduling software had no significant effects on psychological distress, self-rated health and work-life conflict.

In subgroup analyses, using participatory working time scheduling software was associated with higher control over scheduling of shifts in both employees aged < 50 years and those aged ≥ 50 years. The association of using this software with lower prevalence of short sleep reached statistical significance (*p* <0.05) only in employees aged <50 years. Using the software was not significantly associated with poor workability in younger or older employees.

In a sensitivity analysis, excluding 84 employees who slept 9 h or longer did not change the RR for short sleep in total sample (RR 0.71, 95% CI 0.52–0.97). However, the RR for employees aged <50 years did not remain statistically significant (RR 0.65, 95% CI 0.42–1.01). In another sensitivity analysis, subgroup balancing propensity scores were estimated for employees aged <50 years and those aged >50 years. Again, using the software had beneficial effects on control over scheduling of shifts both in employees aged <50 years and those aged >50 years (Table A2). The association of using the software with lower prevalence of perceived poor workability and short sleep was found only for employees aged <50 years, but the estimates did not reach statistical significance.

## 4. Discussion

In the present study, we used propensity score weighting to balance baseline characteristics between users (N = 881) and non-users (N = 1144) of participatory working time scheduling software and conduct an unconfounded comparison of the two mainly women groups for the risk of low control over scheduling of shifts, short sleep, poor well-being and poor workability at follow-up. Our findings suggested that the use of participatory working time scheduling software increases hospital employees’ control over scheduling of shifts and might reduce the risk of short sleep and poor workability. No association was observed for psychological distress, self-rated health, or work-life conflict.

The beneficial effects of using participatory working time scheduling software on short sleep and workability are plausible. Using self-scheduling software improves employees’ control over working time, and self-control over working time can reduce the risk of short sleep if work shifts that are optimal for individual sleep needs are selected. Shorter sleep is further linked to poor workability [40], and sufficient sleep can improve well-being and productivity [41]. Previous prospective cohort studies found that low worktime control increases the risk of psychological distress [15] and depressive symptoms [17,18]. Improvement of psychological distress could also directly improve sleep. Work-life imbalance has partially mediated the relation between worktime control and depressive symptoms [42]. However, a quasi-experiment showed that worktime self-scheduling via a computer program improved employees’ control over scheduling of working hours, but did not decrease self-perceived stress [24]. Moreover, an intervention study failed to confirm an association between increase in employees’ control over working hours and sleep quality, however, that study recruited a small group of elderly care workers [25]. In the current study, the self-scheduling software was designed to improve employees’ control over working hours and based on a larger number of employees, self-control over working time was associated with a reduced risk of short sleep and workability.

As strengths of the current study, we used propensity score weighting to investigate effects of a working time intervention on employees’ well-being and workability. The study collected data on a large set of confounding factors, sample size was relatively large and the time span of using the software was long enough. There was a small to medium clustering effect [43] and intraclass correlation coefficient for hospital wards was above 0.05 for three of the six study outcomes, being 0.206 for control over scheduling of shifts, 0.082 for self-rated health and 0.051 for work conflict. Due to unmeasured cluster-level confounders, multilevel data structure should be considered in estimating propensity score and/or in weighting analysis [44], and ignoring the multilevel data structure in the analysis can lead to biased estimates [44]. To reduce biases, we used multilevel model to estimate the propensity score.

The study had some limitations. Statistical power was limited, particularly for age-specific subgroup propensity score weighting. The absence of the associations of using the participatory working time scheduling software with workability and short sleep in age-specific propensity score weighting might be due to low statistical power and imbalance in baseline characteristics between users and non-users rather than due to the absence of an association. Some baseline characteristics such as presence of chronic medical conditions were measured in year 2015, but not in year 2017. Although for 2017–2019 cohort we utilised the data on these characteristics measured in year 2015, some exposed and unexposed individuals may have been misclassified. Even though we estimated propensity score using a large set of baseline characteristics, unmeasured and residual confounding cannot be ruled out in observational studies [45]. The propensity score weighting controls partly for unmeasured characteristics that are correlated with measured characteristics.

## 5. Conclusions

This pseudo-experiment adds to previous research on the effect of worktime control interventions on hospital employees’ well-being. Our findings suggest that participatory working time software might provide a practical tool to increase employees’ perceived control over shift scheduling and improve sleep and workability. However, randomised controlled studies are needed to confirm the findings and examine the generalisability of the software across other occupational sectors.

## Figures and Tables

**Figure 1 healthcare-09-01385-f001:**
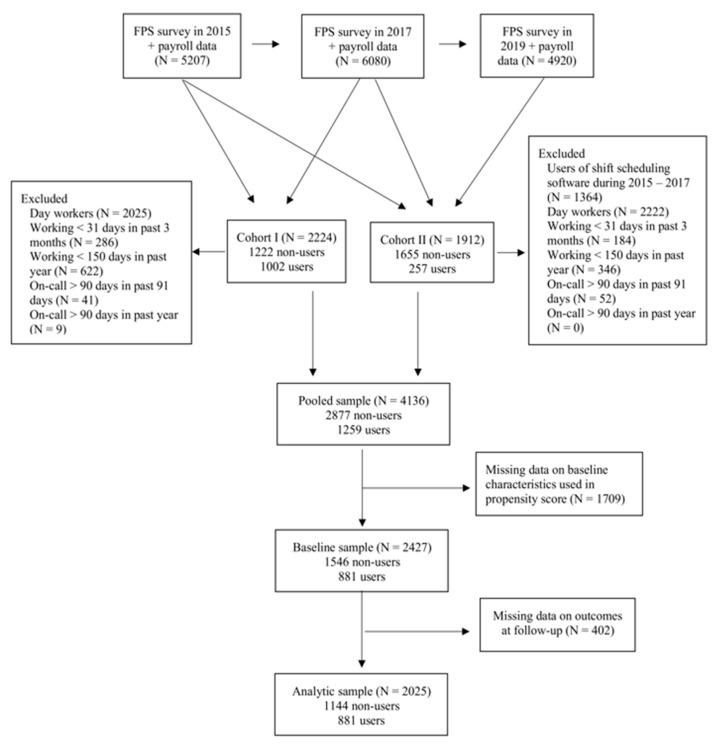
Flow chart of the study population, the Working Hours in the Finnish Public Sector (FPS) study.

**Table 1 healthcare-09-01385-t001:** The baseline characteristics of the study population by age groups.

Characteristic	<50 Years (1499)		≥50 Years (N = 928)		All (N = 2427)	
	N	%	N	%	N	%
Female sex	1330	88.7	849	91.5	2179	89.8
Education						
Primary or civic school	2	0.1	100	10.8	102	4.2
Middle or comprehensive school	530	35.4	440	47.4	970	40.0
Matriculation examination certificate	967	64.5	388	41.8	1355	55.8
Types of usual shift						
Shift work without night shifts (two-shift work)	392	26.1	381	41.1	773	31.8
Shift work with night work (three-shift work)	1019	68.0	464	50.0	1483	61.1
Regular night work	34	2.3	29	3.1	63	2.6
Other irregular work	54	3.6	54	5.8	108	4.5
Smoking						
Past	401	26.8	224	24.1	625	25.8
Current	240	16.0	112	12.1	352	14.5
Body mass index						
Overweight	468	31.2	359	38.7	827	34.1
Obese	295	19.7	225	24.3	520	21.4
Job strain	518	34.6	358	38.6	876	36.1
Duration of sleep (hours)						
Short (≤6)	171	11.4	181	19.5	352	14.5
Long (≥9)	60	4.0	24	2.6	84	3.5
Control over scheduling of shifts						
Intermediate	608	40.6	383	41.3	991	40.8
Low	281	18.8	271	29.2	552	22.7
Poor perceived workability	106	7.1	145	15.6	251	10.3
Poor self-rated health	23	1.5	33	3.6	56	2.3
Work-life conflict	677	45.2	384	41.4	1061	43.7
Psychological distress (GHQ12 score ≥ 3)	449	30.0	262	28.2	711	29.3

**Table 2 healthcare-09-01385-t002:** The prevalence of the study outcomes at follow-up by age groups.

Outcome	<50 Years (N = 1245)		≥50 Years (N = 780)		All (N = 2025)	
	N	%	N	%	N	%
Control over scheduling of shifts						
High	592	47.6	251	32.5	843	41.8
Intermediate	436	35.1	325	42.0	761	37.8
Low	215	17.3	197	25.5	412	20.4
Perceived workability						
Poor	129	10.4	127	16.3	256	12.7
Moderate or high	1109	89.6	650	83.7	1759	87.3
Self-rated health						
Poor	44	3.5	44	5.7	88	4.4
Moderate or good	1198	96.5	732	94.3	1930	95.6
Work-life conflict						
Yes	642	51.7	362	46.5	1004	49.7
No	599	48.3	417	53.5	1016	50.3
Psychological distress						
Yes	440	36.4	223	29.7	663	33.9
No	768	63.6	527	70.3	1295	66.1
Sleep per night						
≤6 h	130	10.5	128	16.5	258	12.8
>6 h	1110	89.5	648	83.5	1758	87.2

**Table 3 healthcare-09-01385-t003:** Comparison of the baseline characteristics in the unweighted and weighted samples.

Characteristic	Unweighted				Weighted			
	Users (%)	Non-Users (%)	Standardised Difference (%)	*p*	Users (%)	Non-Users (%)	Standardised Difference (%)	*p*
Female sex	89.2	90.1	−2.9	0.48	89.2	90.6	–4.4	0.28
Age (years), mean	43.3	45.0	−15.8	< 0.001	43.3	43.4	−0.7	0.86
Age ≥ 50	34.8	40.2	−11.0	0.009	34.8	34.9	−0.2	0.95
Education								
Middle or comprehensive school	30.5	45.3	−30.9	< 0.001	30.5	41.8	−23.4	< 0.001
Matriculation examination certificate	67.4	49.2	37.5	< 0.001	67.4	54.3	27.0	< 0.001
Part-time job	6.4	11.8	−18.9	< 0.001	6.4	9.4	−10.5	0.010
Being a supervisor	2.3	3.0	−4.8	0.26	2.3	2.4	−0.5	0.89
Types of usual shift								
Shift work with night work (three-shift work)	75.0	53.2	46.8	< 0.001	75.0	64.6	22.3	< 0.001
Other shifts	6.0	7.6	−6.5	0.13	6.0	6.1	−0.6	0.96
Current smoking	14.6	14.4	0.6	0.88	14.6	13.5	3.3	0.42
Alcohol consumption	46.3	42.9	6.9	0.10	46.3	44.2	4.2	0.32
Leisure time physical activity								
Moderate	29.9	33.6	−8.1	0.055	29.9	30.0	−0.3	0.95
Strenuous	37.1	33.1	8.4	0.046	37.1	38.1	−2.1	0.62
Body mass index, mean	26.2	26.9	−13.4	0.002	26.2	26.4	−4.1	0.32
Job demands	65.0	64.1	2.0	0.64	65.0	65.5	−0.9	0.82
Job strain	31.9	38.5	−13.8	0.001	31.9	35.3	−7.2	0.087
Procedural justice								
Intermediate	38.8	35.3	7.4	0.079	38.8	36.8	4.1	0.32
Low	41.7	35.2	13.3	0.002	41.7	37.1	9.4	0.026
Control over scheduling of shifts								
Intermediate	39.4	41.7	−4.6	0.27	39.4	42.0	−5.3	0.20
Low	16.2	26.5	−25.1	< 0.001	16.2	21.2	−12.1	0.003
Poor perceived workability	9.3	10.9	−5.4	0.20	9.3	9.4	−0.3	0.95
Poor self-rated health	2.2	2.4	−1.6	0.70	2.2	2.4	−1.5	0.73
Work-life conflict	45.3	42.8	5.0	0.23	45.3	44.2	2.2	0.60
Psychological distress	29.5	29.2	0.7	0.86	29.5	28.2	2.8	0.50
Duration of sleep (hours)								
Short (≤6)	13.1	15.3	−6.5	0.12	13.1	14.4	−3.9	0.34
Long (≥9)	4.5	2.8	9.0	0.028	4.5	3.4	5.9	0.17
Chronic diseases								
Allergy	32.7	28.2	9.8	0.020	32.7	30.6	4.5	0.29
Bronchitis	2.2	3.9	−10.1	0.021	2.2	3.2	−6.0	0.14
Hypertension	15.9	21.3	−13.9	0.001	15.9	17.8	−5.0	0.22
Rheumatoid arthritis	1.4	1.8	−3.1	0.46	1.4	1.3	0.3	0.93
Low back pain or sciatica	17.3	17.2	0.3	0.94	17.3	17.8	−1.3	0.75
Migraine	20.9	17.5	8.7	0.038	20.9	21.9	−2.7	0.54
Mental disorders	11.8	10.3	4.8	0.24	11.8	10.8	3.1	0.46
Diabetes	2.6	3.1	−3.0	0.48	2.6	2.8	−1.2	0.76
Sleep apnea	0.8	2.1	−10.7	0.016	0.8	1.5	−6.0	0.127
Changes at work								
Medium	44.7	40.2	9.1	0.031	44.7	45.1	−0.8	0.85
Large	39.5	41.8	−4.7	0.27	39.5	39.5	0.1	0.98
Employee’s influence on changes at work								
Small	47.7	45.7	4.0	0.34	47.7	48.0	−0.6	0.87
None	49.3	51.7	−4.8	0.25	49.3	49.7	−0.9	0.83

**Table 4 healthcare-09-01385-t004:** Effects of using participatory working time scheduling software on the outcomes by age groups. RR, risk ratio.

Outcome	<50 Years				≥50 Years				All			
	Non-Users	Users	RR	95% CI	Non-Users	Users	RR	95% CI	Non-Users	Users	RR	95% CI
Control over scheduling of shifts												
Good	242	350	1		121	130	1		363	480	1	
Intermediate or low	428	223	0.70	0.53–0.91	347	175	0.79	0.69–0.91	775	398	0.72	0.60–0.85
Control over scheduling of shifts												
Good or intermediate	513	515	1		311	265	1		824	780	1	
Low	157	58	0.30	0.20–0.44	157	40	0.44	0.29–0.69	314	98	0.34	0.25–0.46
Perceived workability												
Good	595	514	1		388	262	1		983	776	1	
Poor	74	55	0.73	0.48–1.10	82	45	0.84	0.56–1.25	156	100	0.74	0.55–0.99
Perceived health												
Good	645	553	1		445	287	1		1090	840	1	
Poor	24	20	0.72	0.39–1.33	27	17	1.21	0.51–2.91	51	37	0.88	0.51–1.52
Work-life conflict												
No	327	272	1		259	158	1		586	430	1	
Yes	342	300	0.95	0.76–1.19	214	148	1.18	0.98–1.43	556	448	1.03	0.87–1.23
Psychological distress												
No	412	356	1		315	212	1		727	568	1	
Yes	240	200	0.79	0.58–1.06	138	85	0.87	0.66–1.14	378	285	0.83	0.66–1.05
Short sleep (≤6 h)												
No	587	523	1		388	260	1		975	783	1	
Yes	81	49	0.65	0.42–0.99	82	46	0.86	0.58–1.26	163	95	0.70	0.52–0.95

## Data Availability

The data presented in this study are available on request from the authors.

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
