# Peer review of "The Effect of Using Participatory Working Time Scheduling Software on Employee Well-Being and Workability: A Cohort Study Analysed as a Pseudo-Experiment"

_healthcare, 2021, doi:10.3390/healthcare9101385_

Round 1

Reviewer 1 Report

Overall the paper is well written. but please fix my follow comments:

  1. please give some background info in the abstract
  2. the data are purely from hospitals, I suggest the authors add some background information in the introduction about the problems in the hospitals. 
  3. please define pseudo-experiment 
  4. please have a strengthened literature review part on why scheduling of shifts, perceived workablity, self-rated health,  psychological distress, short sleep, and work-life conflict were used in the paper, corresponding to lines 95&96
  5. line 187, what is intraclass correlation? and what the correlation means? for example, what the following sentence means " Intraclass correlation coefficient was 0.206 for control over scheduling of shifts"?

Author Response

Overall the paper is well written. but please fix my follow comments:

Response: Thank you for your assessment.

please give some background info in the abstract.

Response: We have added background information to the abstract.

the data are purely from hospitals, I suggest the authors add some background information in the introduction about the problems in the hospitals. 

Response: The study recruited shift workers, and the most common workplaces for shift work in healthcare are hospitals. We have described the unfavourable effects of shift work in the introduction on page 1, which apply to all shift workers.

please define pseudo-experiment 

Response: We have defined a pseudo-experiment in the introduction as “a study intended to assess causal relationship without random assignment” on page 2.

please have a strengthened literature review part on why scheduling of shifts, perceived work ability, self-rated health, psychological distress, short sleep, and work-life conflict were used in the paper, corresponding to lines 95&96.

Response: The outcomes of the study were defined before initiation of the intervention. We have added this information on lines 98-99 “This pseudo-experiment was registered on ClinicalTrials.gov (NCT02775331) before initiation of the intervention”.  

The aim of this study was to explore whether participatory working time scheduling software improves shift workers’ well-being and perceived workability. Self-rated health, psychological distress, short sleep, and work-life conflict were used as indicators of well-being. In addition, we have described in the introduction some earlier literature on worktime control and the main effects of shift work on health and wellbeing that were the main reasons for making the original hypothesis that participatory working time scheduling (improving worktime control) could influence perceived health and wellbeing.

line 187, what is intraclass correlation? and what the correlation means? for example, what the following sentence means " Intraclass correlation coefficient was 0.206 for control over scheduling of shifts"?

Response: Intraclass correlation coefficient measures similarities of responses within clusters. High intraclass correlation coefficient reduces effective sample size, and a larger sample is needed when there is correlation between individuals within the same cluster. In the example, intraclass correlation of 0.206 indicates that participants within clusters of wards had similar perceptions on their ability to control their shift schedule. We have explained intraclass correlation under Statistical analysis as follows:

“Intraclass correlation coefficient measures similarities of responses within clusters” on page 5.

Reviewer 2 Report

  1. Resesrch gap literature review in the introduction should be addressed.
  2. Which theory is used in your paper need to explain.
  3. What is your contribution need to explain.
  4. English need improve throughout the whole paper. 
  5. Reference need to update. 

Author Response

Resesrch gap literature review in the introduction should be addressed.

Response: In the revised introduction we have summarized the findings of the limited number of published controlled trials as well as prospective cohort studies on the health effects of worktime control and self-scheduling of shifts. As the rationale of the current study, we have stated that there is an increased need for good-quality randomized controlled trials and pseudo-trials to identify practical ways to reduce the unfavourable effects of shift work, a major research gap on page 2.

Which theory is used in your paper need to explain.

Response: The current study is a pseudo-experiment, testing the hypothesis that improving work-time control with participatory shift scheduling can improve perceived health and well-being. We have now clarified the theoretical background of the study in the introduction. Theoretical groundings for the assumed benefits of increased work-time control stem from several motivational and occupational health theories. According to concepts, such as the demand-control model, this should be the case. References to this theory have been added.

What is your contribution need to explain.

Response: In the revised abstract, we note that “Shift workers are at increased risk of health problems. Effective preventive measures are needed to reduce the unfavorable effects of shift work” (page 1). We have addressed this limitation and reported throughout the manuscript the new findings of the study. The abstract concludes that “the use of participatory working time scheduling software was associated with increased employees’ perceived control over scheduling of shifts and improved sleep and self-rated workability.” (page 1). In the discussion, we have noted that “Our findings suggest that participatory working time software might provide a practical tool to increase employees’ perceived control over shift scheduling and improve sleep and workability. However, randomized controlled studies are needed to confirm the findings and examine the generalizability of the software across other occupational sectors.” (page 11).

English need improve throughout the whole paper. 

Response: We have checked the manuscript and corrected linguistic errors.

Reference need to update. 

Response: The reference list is now updated. We have added a reference to Demand-control theory, but no new empirical study has been published. There are very few earlier studies on participatory working time scheduling, and they have been cited. The older references also include several references to the original scales used in the Finnish Public Sector study since 1990´s, e.g., Blaxter 1987, Dohrenwend et al. 1978, and Mårdberg et al 1991.  

Reviewer 3 Report

The study addresses a topic of interest given that a significant number
of workers are subjected to shift work systems.
The methodology and presentation of results is adequate and only a minor change or suggesstion are proposed
Introduction: good justification of the interest of the study. Methods:
1- clarify if the final sample of 2025 employees, 881 were users of the software in the baseline or in the follow-up study or in both. 2- Could you describe the professional category of the participants? 3- Have you ever wondered how long you have been using the software? 4- Incorporate the references or validation studies on which the workability measures are based,. self-rated health; sleep duration and work-life conflict 5- Clarify paragraph 145-146. for 2017-2019 cohrt we utilized data collected in 2015 survey. Are they the same or a different cohort sample? Outcome: Please include the total "N" values ​​in the top row of table 2, as it appears in table 1. Report those missing values ​​for some of the outcomes described (example control over scheduling).

Author Response

The study addresses a topic of interest given that a significant number of workers are subjected to shift work systems. The methodology and presentation of results is adequate and only a minor change or suggestion are proposed

Response: Thank you for your assessment.

Introduction: good justification of the interest of the study.

Response: Thank you for your positive comments.

Methods:
1- clarify if the final sample of 2025 employees, 881 were users of the software in the baseline or in the follow-up study or in both.

Response: The final sample consisted of participants with no missing data on the baseline characteristics and the outcomes of interest. We have clarified this issue on lines 97-98.

2- Could you describe the professional category of the participants?

Response: We have listed the job titles with at least 50 participants in the results section on page 6 as follows:

“The job titles with 50 or more participants included nurse (36.2%), practical nurse (9.5%), department secretary (6.8%), hospital nurse (4.3%), radiology nurse (4.1%), midwife (3.5%), specialist (2.5%) and mental health nurse (2.1%).”

3- Have you ever wondered how long you have been using the software?

Response: We have applied intention-to-treat principle and included all participants who used the software for a month or longer. We have clarified this on page 4, lines 144-146.

4- Incorporate the references or validation studies on which the workability measures are based, self-rated health; sleep duration and work-life conflict

Response: These measures are commonly used in surveys and we have revised the Outcome section and added the relevant references on page 3.

5- Clarify paragraph 145-146. for 2017-2019 cohort we utilized data collected in 2015 survey. Are they the same or a different cohort sample?

Response: Data on the characteristics of both 2015-2017 and 2017-2019 cohorts were collected in 2015 and 2017. For 2015-2017 cohort we used 2015 measures as the baseline of the study and for 2017-2019 cohorts we used 2017 measures as the baseline measures. In the absence of information in 2017 for 2017-2019 cohort we used 2015 measures as the baseline of the study.

Outcome: Please include the total "N" values ​​in the top row of table 2, as it appears in table 1. Report those missing values ​​for some of the outcomes described (example control over scheduling).

Response: We have added the number of participants to column headings of Table 2. Missing data are reported in the methods and Figure 1.

Round 2

Reviewer 2 Report

I suggest that you adopt appropriate academic expression. I am concerned that the grammar and spelling needs to be checked.